# Fast Inference of Removal-Based Node Influence

Submission Id: 434*

## ABSTRACT

Graph neural networks (GNNs) have been widely utilized to capture the underlying information propagation patterns in graph-structured data. While remarkable performance has been achieved in extensive classification tasks, there comes a new trending topic of identifying influential nodes on graphs. This paper investigates a new yet practical problem of evaluating the influence of node existence itself, which aims to efficiently measure the overall changes in the outputs of a trained GNN model caused by removing a node. A realistic example is, "Under a task of predicting Twitter accounts' polarity, had a particular account not appeared, how might others' polarity be changed?". A straightforward way to obtain the node influence is to alternately calculate the influence of removing each node, which is reliable but time-consuming. The related lines of work, such as graph adversarial attack and counterfactual explanation, cannot directly satisfy our needs since they typically suffer from low efficiency on large graphs. Besides, they cannot individually evaluate the removal influence of each node. To upgrade the efficiency, we design an efficient algorithm, **NO**de-**R**emoval-based f**A**st GNN inference (**NORA**), which uses the gradient of the neural networks to approximate the node-removal results. It only costs one forward propagation and one backpropagation to approximate the influence score for all nodes. Extensive experiments are conducted on six benchmark datasets, where NORA exceeds the compared methods. Our code is available at https://anonymous.4open.science/r/NORA.

## CCS CONCEPTS

• **Mathematics of computing** → **Graph algorithms**; • **Information systems** → *Social networks*.

## KEYWORDS

node influence evaluation, graph neural network, network analysis

## 1 INTRODUCTION

In recent years, the booming development of big data has brought about many relational data, that can be naturally represented as graphs. Evaluating node influence and identifying influential nodes on a graph has become a trending and beneficial topic [14]. It can help with viral advertising [9, 24, 37], online news dissemination [11, 25], police breaking down a criminal network [7], pandemic control [13, 54], etc. A lot of research on the "influence maximization" problem [16, 18, 23, 26, 28, 29, 44, 48, 50, 53, 62] focus on identifying influential nodes whose triggered influence spreading range can be maximized. These works can answer the question: "Which Twitter accounts post information that can spread to the greatest amount of audiences?"

Yet, another question is under-explored: "If a Twitter account had never appeared, how could other Twitter users' opinions/interactions (e.g. following, retweeting, and replying) have been?", such as the example we illustrate in Figure 1. Actually, studying the influence

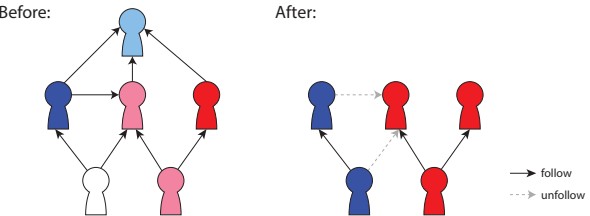

**Figure 1: An example of a possible node-removal scenario on a social network. Red versus Blue represents two different opinions. Color shade represents how firm a user's stance is.**

of node removal can benefit many real-world applications including finding the bottlenecks and improving the infrastructure network robustness [6, 27], modeling how vaccination can decrease virus spreading [3, 13, 54] and figuring out the top scientists contribute to knowledge spreading based on a science co-authorship network [1, 3, 22]. A lot of research on the "network dismantling" problem[30, 35, 38–40, 58, 63] have studied the structural influence of node removal. However, the task-specific influence of node removal considering both attributes and structures has been under-explored. Therefore, we focus on measuring the influence of node existence itself by evaluating the task-specific influence of node removal.

Graph neural networks (GNNs) are among the most powerful graph representation learning tools. Different from research on the "influence maximization" problem that uses a propagation model to simulate node influence spreading range, we use GNNs as a surrogate to capture the information propagation patterns. Propagation models cannot evaluate the influence of node removal, but it is not the case for GNNs. Based on the message-passing nature of GNN [10], we assume that a trained GNN model can capture the propagation patterns of a graph. After removing a node, we can use a pre-trained GNN's new outputs to simulate the scenario if that removed node had not existed based on the learned propagation patterns. For node classification task, it simulates what other node labels could have been; for link prediction task, what the connections could have been; for graph classification task, what the graph label could have been. We calculate the influence of node removal as the total variation distance between the original outputs and new outputs of the trained GNN model, which is illustrated in Figure 2.

We aim to calculate the influence score for each node. However, brutal-force direct calculation is very time-consuming, so we demand an efficient method. Our method that changes GNN predictions by changing its input graph structure is similar to some common practice in graph adversarial attack and graph counterfactual explanation, though we aim at a different problem setting. Graph adversarial attack aims to maximally undermine GNN model performance or change GNN predictions by inducing unnoticeable perturbations. The perturbations mainly include modifying node

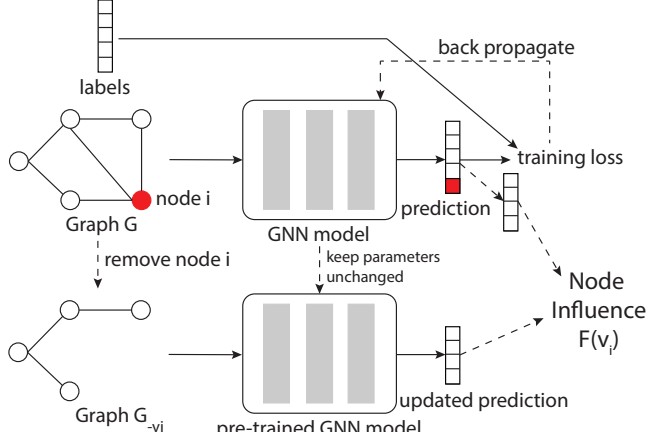

**Figure 2: Our schema of evaluating node influence based on node removal. The GNN model is trained on the original graph, then we remove a node and apply the trained GNN to the new graph structure. We measure the total variation distance between the originally predicted and newly predicted distributions of node/edge/graph classes.**

features [33, 67, 69], injecting new nodes [5, 8, 19, 45, 47, 52, 66], or modifying edges [51, 57, 64, 68]. To the best of our knowledge, none of the adversarial attack methods utilizes node removal, since it is impractical in real-world attacking scenarios.

Graph counterfactual explanation aims to explain a GNN's prediction of a target node/edge/graph by finding the minimum perturbation on the input graph that can change the prediction of the target [43]. They do utilize node removal [17, 34, 41, 46, 55, 60, 61]. However, directly using these methods to evaluate node influence confronts two difficulties. First, we evaluate the influence of removing a particular node on all other nodes/edges, while graph counterfactual explanation evaluates the influence of removing many nodes and edges on a single target. Second, graph counterfactual explanation strategies are not typically good at scaling up to handle large graphs. Most of the existing graph-level classification datasets contain a lot of small graphs, such as molecules. For node classification and link prediction tasks, some models only need to consider the computation graph of a targeted node/edge, which is also small. Most of the existing works mentioned above only conduct experiments on graphs with less than 4,000 nodes. LARA [41] designs a scalable model to predict the influence of surrounding nodes on the target node, but it requires the time consuming labeling of the ground truth, thus it is efficient in space but not in time. Our method is much faster.

The node influence measurement problem we proposed has not been studied yet, and related lines of work cannot directly satisfy our demands. To efficiently calculate the node influence score, we use the gradient to approximate the influence based on the first-order derivatives and heuristics. We propose the algorithm, **NO**de-**R**emoval-based f**A**st GNN Inference (**NORA**), that only needs one forward propagation and one backpropagation to approximate the removal influence for all nodes. Since we are studying a new problem without mature baselines, we adapt two approaches in graph

counterfactual explanation as supplementary baselines to this problem. We conduct extensive experiments on six datasets. The experimental results demonstrate the effectiveness and efficiency of NORA. To sum up, this paper makes the following contributions:

- We propose a novel perspective of evaluating node influence based on node removal and a pre-trained GNN.
- We propose an efficient and effective algorithm, NORA, to approximate the removal influence for all nodes.
- Experimental results on six datasets demonstrate that NORA outperforms the baselines on performance and efficiency.

## 2 RELATED WORK

### 2.1 Graph Adversarial Attack

Graph adversarial attack aims to undermine GNN performance or change GNN predictions by imposing a small perturbation to the graph within a limited budget. Zügner et al. [67, 69] started the race of graph adversarial attacks. Pioneering works are mainly based on modifying node features [33, 67, 69] and perturbing edges [51, 57, 64, 68], including adding, removing, and rewiring edges. Some recent works [5, 8, 19, 20, 45, 47, 52, 66] study the node injection attack, which injects some nodes into the graph and connects them with some existing nodes. Among them, Chen et al. [5] prove that the node injection attack can theoretically cause more damage than the graph modification attack with less or equal modification budget. G-NIA model [47] sets a strong limitation that the attacker can only inject one node with one edge, and it achieves more than 90% successful rate in the single-target attack on Reddit and ogbn-products datasets. They demonstrate the strong potential of altering nodes' existence. To the best of our knowledge, none of the adversarial attack methods considers node removal, since it is impractical in real-world applications. Nonetheless, as our target is to analyze node influence instead of attacking, node removal is worth exploring.

### 2.2 Graph Counterfactual Explanation

Graph counterfactual explanation explains why a GNN model gives a particular result. Such as, to explain the GNN prediction of a target node in the node classification task, a target edge in the link prediction task, or a target graph in the graph classification task. The explanation is provided by finding the minimum perturbation on the input graph that can change the prediction of the target. There are some methods [2, 31, 32, 59] based purely on edge removal. Some methods utilize both node removal and edge removal by optimizing mask matrices [46, 55], predicting node influence with neural network [41], applying graph generation models [34, 60], or searching for an optimal neighbor graph [17, 61]. As analyzed in Section 1, these methods are not directly applicable to evaluating the proposed node influence, so we adapt two famous methods as supplementary baselines to this novel problem. CF-GNNExplainer [31] optimizes a real-value mask matrix that multiplies the adjacency matrix during training, and elements in the mask matrix must be within range [0, 1]. During inference, elements below 0.5 indicate edge removal. We adapt it to also consider node removal with a node mask matrix. Optimizing the mask matrix is a very common practice [2, 46, 55, 59], so we use CF-GNNExplainer as a baseline. As discussed in Section 1, most graph explanation methods are

not scalable. To solve the problem, a recent work, LARA [41], uses a GNN to predict node influence, whose parameter size does not grow with the input graph size. We adapt it as our second baseline.

### 2.3 Network Dismantling

Network dismantling studies the structural influence of node removal on unattributed graphs. It aims to maximally decrease network connectivity by analyzing by removing influential nodes. The influence is usually evaluated by the network connectivity, such as the size of the largest connected component, efficiency (i.e. the average of the reciprocals of shortest path lengths of all node pairs), etc [36]. Betweenness centrality is one of the most widely-used methods in the network dismantling problem setting to measure node influence [30, 35, 38, 39, 58]. It is the ratio of shortest paths that pass through a node among all shortest paths between all node pairs. We use it as a simple baseline in our experiments.

## 3 PROBLEM DEFINITION

### 3.1 Notations

A graph $G = (V, E)$ consists of $N$ nodes $V = \{v_1, v_2, ..., v_N\}$ and edges $E = \{e_{ij} | j \in \mathcal{N}(i)\}$, where $\mathcal{N}(i)$ denotes the neighbor nodes of $v_i$ (without $v_i$), and $e_{ij}$ denotes the edge from $v_i$ to $v_j$. $\hat{\mathcal{N}}(i)$ denotes neighbor nodes of $v_i$ plus $v_i$ itself. $A$ denotes the adjacency matrix. Each node $v_i$ is associated with a feature vector $x_i \in \mathbb{R}^d$, and a label $y_i \in \mathbb{R}$ if node classification task is applicable. We denote the degree of $v_i$ as $d_i = |\mathcal{N}(i)|$. When we remove node $v_r$ ($r \in \{1, 2, \ldots, N\}$) and all edges connected with $v_r$, from graph $G$, we denote the new graph as $G_{-v_r}$. $g_\theta$ denotes a trained GNN model. We denote as $v_r$ the target node to analyze removal influence, and $\text{F}_{g_\theta}(v_r)$ denotes the proposed influence.

Graph neural networks (GNNs) generally follow the message-passing framework [10]. A GNN model consists of multiple graph convolutional layers. In a typical graph convolutional layer, a node updates its representation by aggregating its neighbor nodes' representations:

$$h_i^{(l)} = U_l(h_i^{(l-1)}, \text{AGG}(\sum_{j \in \mathcal{N}(i)} \text{MSG}_l(h_j^{(l-1)}, h_i^{(l-1)}))), \quad (1)$$

where $h_i^{(l)}$ denotes the node $v_i$'s representation after passing the $l$-th layer ($l \in 1, 2, \ldots$), and $h_i^{(0)}$ denotes the input features. $\text{MSG}_l$ is the message function, AGG is the aggregation function, and $U_l$ is the update function.

### 3.2 Problem Definition

In order to evaluate the change of node removal, we use GNN models as a surrogate to predict the scenario where the removed node does not exist based on the existing propagation patterns. GNN models typically make prediction on a node label, edge existence, or graph class, via transforming its output to a vector, which denotes the probability of different classes or options. In general, we measure the change of prediction by the $\ell_1$-norm of the difference between the original and updated probability vectors. The difference can equally capture the prediction change for every class. For graph classification, we directly use the prediction change. For

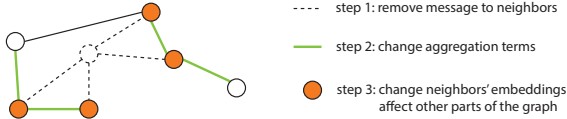

**Figure 3: The influence calculation decomposition of our method.**

node classification and link prediction, we evaluate the sum of the prediction change of all remaining nodes/edges.

**Definition 1. (Node Influence in Node Classification Task)** Given a node classification model $g_\theta$ trained on graph $G$, we denote its prediction of node $v_i$ as $g_\theta(G)_i$. The influence of removing node $v_r \in V$ is calculated as:

$$\text{F}_{g_\theta}(v_r) = \sum_{i=1, i \neq r}^{N} ||f(g_\theta(G)_i) - f(g_\theta(G_{-v_r})_i)||_1, \quad (2)$$

where $f(\cdot)$ is the optional MLP layers and softmax layer that transform a GNN's output to the probabilistic vector.

**Definition 2. (Node Influence in Link Prediction Task)** Given a link prediction model $g_\theta$ trained on graph $G$, we denote its prediction of edge $e_{ij}$ as $g_\theta(G)_{e_{ij}}$. We use $D_e$ to denote the whole link prediction set, and $D_r$ to denote edges that link $v_r$. The influence of removing node $v_r \in V$ is calculated as:

$$\text{F}_{g_\theta}(v_r) = \sum_{e_{ij} \in D_e - D_r} ||f(g_\theta(G)_{e_{ij}}) - f(g_\theta(G_{-v_r})_{e_{ij}})||_1, \quad (3)$$

The definition can be similarly generalized to the graph classification task, where we simply take $||f(g_\theta(G)) - f(g_\theta(G_{-v_r}))||_1$.

The ground truth is generated by the brute-force algorithm, where we alternatively remove each node from the original graph one at a time, and calculate the node influence. Iterating through all nodes causes short efficiency. One intuitive way for acceleration is neighborhood sampling. If the GNN has $l$ layers, removing a node will only affect the outputs of its $l$-hop neighborhood, and computing their new outputs will only require a $2l$-hop neighborhood. However, it is still time-consuming, especially on dense-connected graphs, (e.g., ogbn-arxiv and two Twitter datasets in our experiments) where $2l$-hop neighborhoods might already contain most of the nodes. Therefore, we need to look for an efficient and effective method to calculate the influence score.

## 4 METHODS

To upgrade the efficiency, we propose **No**de-**R**emoval-based F**a**st GNN Inference (**NORA**) algorithm. In general, we approximate the influence of the single-node-removal process by decomposing the calculation process into three parts, which correspond to three parts of changes caused by the node removal. Figure 3 illustrates the three parts. We will describe our approximation algorithm in detail in the following subsections.

### 4.1 Influence Score Calculation Decomposition

The general idea of NORA is that we approximate the influence of node removal via first-order derivatives. We only need the gradient information from one backpropagation to approximately calculate

the influence scores for all nodes.Our method could be applied to node classification and link prediction tasks, and can also be extended to the graph classification task. In general, our method could be adapted to various downstream tasks of a GNN model.

Equation 1 illustrates a message-passing GNN layer. A typical parameterization of it is:

$$h_i^{(l)} = \sigma(W_u^{(l)}(W_s^{(l)} h_i^{(l-1)} + \sum_{j \in N(i)} \alpha_{ji} W_m^{(l)} h_j^{(l-1)})), \quad (4)$$

where $\sigma$ denotes the activation function, $W_u^{(l)}$, $W_s^{(l)}$, and $W_m^{(l)}$ are model parameters. $\alpha_{ji}$ is the edge normalization of messages coming from $v_i$'s neighbors and is usually related to node degree or attention mechanism, e.g., $\alpha_{ji} = 1/\sqrt{|N(i)||N(j)|}$ in GCN [21]. Suppose the GNN model has $L$ layers, the last layer output $g_\theta(G)_i = h_i^{(L)} \in R^c$ is the predicted class probability, where $c$ is the number of classes.

We cannot directly calculate the first-order derivatives based on Equation 2, since there is a 1-norm. However, intuitively, removing a node usually causes consistent change to the class of other nodes, e.g., raising the probability of a particular class for all nodes. Therefore, we can rewrite the formula. We denote as $f_r = \sum_{i=1, i \neq r}^N h_i^{(L)}$ the sum of all node predictions except for node $v_r$, and we denote as $\delta f_r$ the change of $f_r$ when removing node $v_r$.

LEMMA 1. *If removing $v_r$ consistently changes the class distribution of other nodes, the influence defined in Equation 2 equals:*

$$|| \sum_{i=1, i \neq r}^N g_\theta(G)_i - \sum_{i=1, i \neq r}^N g_\theta(G_{-v_r})_i ||_1 = ||\delta f_r||_1$$

$$= || \sum_{i \neq r} \delta h_i^{(L)} ||_1 = || \sum_{i \neq r} \frac{\partial f_r}{\partial h_i^{(L)}} \delta h_i^{(L)} ||_1. \quad (5)$$

Though the second line contains the derivative symbol, it is strictly equal because $\frac{\partial f_r}{\partial h_i^{(L)}} = 1$. We write it in this form because we want to keep a uniform form with later formulas. We can extend this form from the last layer's formula to the frontmost layer. Here we analyze how to extend it from the $L$-th layer to the $(L-1)$-th layer. Since the 1-norm is difficult to compute, we first ignore it and just approximate $\delta f_r$.

In a typical GNN layer in Equation 4, the model parameters are fixed during inference, but $\alpha_{ji}$ and $h_j^{(L-1)}$ might change due to removing $v_r$. Therefore, we can approximate $\delta h_i^{(L)}$ with the first-order derivatives:

$$\delta h_i^{(L)} \approx -I(v_r \in N(i)) \frac{\partial h_i^{(L)}}{\partial h_r^{(L-1)}} h_r^{(L-1)}$$

$$+ \sum_{j \in \hat{N}(i), j \neq r} (\frac{\partial h_i^{(L)}}{\partial \alpha_{ji}} \delta \alpha_{ji} + \frac{\partial h_i^{(L)}}{\partial h_j^{(L-1)}} \delta h_j^{(L-1)}), \quad (6)$$

where $I(.)$ is the indicator function. Then by combining the above formula with the definition of $\delta f_r$, we can derive the following formula.

LEMMA 2. *We can approximate $\delta f_r$ for the GNN model described in Equation 4 with a second-order error term as:*

$$\delta f_r \approx - \sum_{i \in N(r)} \frac{\partial f_r}{\partial h_i^{(L)}} \frac{\partial h_i^{(L)}}{\partial h_r^{(L-1)}} h_r^{(L-1)} + \sum_{i \neq r} \sum_{j \in \hat{N}(i), j \neq r}$$

$$(\frac{\partial f_r}{\partial h_i^{(L)}} \frac{\partial h_i^{(L)}}{\partial \alpha_{ji}} \delta \alpha_{ji}) + \sum_{i \neq r} \sum_{j \in \hat{N}(i), j \neq r} (\frac{\partial f_r}{\partial h_i^{(L)}} \frac{\partial h_i^{(L)}}{\partial h_j^{(L-1)}} \delta h_j^{(L-1)}). \quad (7)$$

The error term is in the second order because we are using the first-order derivatives to approximate. We now decompose the calculation into three terms, divided by "+" in the above formula. The first term measures the direct influence of the disappearance of $v_r$'s latent representations, which decreases an input to its neighbor node; The second term measures the change of its neighbor's edge normalization term $\alpha_{ji}$; and the third term measures the change of other nodes' latent representations, which will influence further neighbors. The three terms correspond to the three kinds of influence in Figure 3.

## 4.2 Approximation of Each Decomposed Term

**Term 1: Direct impact to the neighbors.** For clarity, the first term refers to the portion between the first minus sign and the first plus sign in Equation 7. To begin with, by applying the chain rule, the first term equals to:

$$\frac{\partial f_r}{\partial h_r^{(L-1)}} h_r^{(L-1)} - \frac{\partial f_r}{\partial h_r^{(L)}} \frac{\partial h_r^{(L)}}{\partial h_r^{(L-1)}} h_r^{(L-1)}. \quad (8)$$

The derived equation consists of two parts. The form of the first part is simpler and more convenient to handle, so we want to eliminate the second part and only keep the first part. We do this by approximating the ratio of the second part to the first part. Here we make a rough assumption that every node is equal, which means they have the same number of neighbors, the same node representation, and the same gradient. We denote the change of node representation, $\delta h_j^{(L-1)}, \forall j \in V$, as $\delta h$. We denote the gradient coming from a neighbor node as $g$, and the gradient coming from the higher-layer representation of a node itself as $\beta g$. $\beta$ is typically higher than 1, because self-loop and residual connection make the gradient coming from the higher-layer representation of a node itself larger than the gradient from the higher-layer representation of neighbor nodes. Therefore, the first part of Equation 8 is $(d_r + \beta)g\delta h$, and the second part is $\beta g \delta h$. Based on their ratio, and by rewriting the enumeration variable $j$ as $i$, we derive the following equation.

LEMMA 3. *If every node in the graph has equal structures and attributes, the first term of Equation 7 equals:*

$$\frac{d_r}{d_r + \beta} \frac{\partial f_r}{\partial h_r^{(L-1)}} h_r^{(L-1)}. \quad (9)$$

In our experiments, we find that the most effective way of calculating $\frac{\partial f_r}{\partial h_r^{(L-1)}} h_r^{(L-1)}$ is to change it to $||\frac{\partial f_r}{\partial h_r^{(L-1)}} \circ h_r^{(L-1)}||_2$. $\circ$ means element-wise product between the two same-dimensional vectors, and $||.||_2$ means the 2-norm.

**Term 2: Aggregation term change.**. In the second term of Equation 7, $\frac{\partial f_r}{\partial h_i^{(L)}} \frac{\partial h_i^{(L)}}{\partial \alpha_{ji}} = \frac{\partial f_r}{\partial \alpha_{ji}}$. We have tried using $\frac{\partial f_r}{\partial \alpha_{ji}}$ but it didn't perform well, so we only consider approximating $\delta \alpha_{ji}$. Then we analyze $\delta \alpha_{ji}$. Unlike the first term, $\delta \alpha_{ji}$ greatly depends on the design of the specific GNN model. Some GNN models, e.g., GCN [21] and GraphSAGE [12], only use structural information like node degree, while some models, e.g., GAT [49] and DrGCN [65], uses the attention mechanism. To reach a flexible and universally adaptable approximation, we use structural measurement. We consider two widely-used GNNs: GCN [21] and GraphSAGE [12]. The edge normalization of GCN is $\alpha_{ji} = 1/\sqrt{|N(i)||N(j)|}$, and that of Graph-SAGE is $\alpha_{ji} = 1/|N(i)|$. If neither $v_i$ nor $v_j$ is $v_r$'s neighbor, $\alpha_{ji}$ of GCN and GraphSAGE does not change.

If $v_i$ or $v_j$ is a neighbor of $v_r$, we combine the fashion of GCN and GraphSAGE to approximate $\delta \alpha_{ji}$. We denote the degree of node $v_i$ as $d_i = |N(i)|$. Suppose $v_i$ is $v_r$'s neighbor, and $v_j$ is $v_i$'s neighbor, we approximate $\delta \alpha_{ji}$ by $\hat{\delta} \alpha_{ji}$:

$$\hat{\delta}\alpha_{ji} = [k_1(\frac{1}{\sqrt{d_i - 1}} - \frac{1}{\sqrt{d_i}}) + (1 - k_1)(\frac{1}{d_i - 1} - \frac{1}{d_i})]$$
$$[k_2 \frac{1}{\sqrt{d_j}} + (1 - k_2)\frac{1}{d_j}], \qquad (10)$$

where $k_1$ and $k_2$ are hyper-parameters ranging in [0,1]. An interesting intuition is that there exist hyper-parameters $k_1$ and $k_2$ that make $\hat{\delta}\alpha_{ji}$ equal to $\delta \alpha_{ji}$ for GCN. Based on $\hat{\delta}\alpha_{ji}$, we approximate the second term as:

$$\delta Topo_r = \sum_{i \in N(r)} \sum_{j \in N(i)} \hat{\delta}\alpha_{ji}. \qquad (11)$$

**Term 3: Hidden representation change.** Using the chain rule to analyze $\frac{\partial f_r}{\partial h_j^{(L-1)}}$, we can simplify the third term. The third term in Equation 7 equals Equation 12, which can be further equally transformed into Equation 13.

$$\sum_{j \neq r} (\frac{\partial f_r}{\partial h_j^{(L-1)}} - \frac{\partial f_r}{\partial h_r^{(L)}} \frac{\partial h_r^{(L)}}{\partial h_j^{(L-1)}})\delta h_j^{(L-1)} \qquad (12)$$

$$= \sum_{j \neq r} \frac{\partial f_r}{\partial h_j^{(L-1)}} \delta h_j^{(L-1)} - \sum_{j \in N(r)} \frac{\partial f_r}{\partial h_r^{(L)}} \frac{\partial h_r^{(L)}}{\partial h_j^{(L-1)}} \delta h_j^{(L-1)}. \qquad (13)$$

Similar to the simplification process of the first term, here we also arrive at a formula with two parts. The form of the first part is more convenient to handle, and it takes the same form as Equation 5, so we want to eliminate the second part and only keep the first part. We make the same rough assumption that every node is equal. Equation 8 as below. We denote the average node degree as $d$. Using the notations from the simplification process of the first term, we can approximate the first part of the third term (Equation 12) as $(N - 1)(d + \beta)g\delta h$, and the second part as $dg\delta h$. Based on their ratio, and by rewriting the enumeration variable $j$ as $i$, we derive the following equation.

LEMMA 4. *If every node in the graph has equal structures and attributes, the third term of Equation 7 equals:*

$$(\sum_{i \neq r} \frac{\partial f_r}{\partial h_i^{(L-1)}} \delta h_i^{(L-1)})(1 - \frac{d}{(N-1)(d+\beta)}). \qquad (14)$$

We use this equation to approximate the third term. Its algebraic form is similar to Equation 5, so the third term can successfully extend the formula to previous layers.

## 4.3 Combined Derivation

By combining the approximations of three terms together, we get:

$$\delta f_r \approx (\sum_{i \neq r} \frac{\partial f_r}{\partial h_i^{(L-1)}} \delta h_i^{(L-1)})(1 - \frac{d}{(N-1)(d+\beta)}) + \delta topo_r$$
$$- \frac{d_r}{d_r + \beta} ||\frac{\partial f_r}{\partial h_r^{(L-1)}} \circ h_r^{(L-1)}||_2. \qquad (15)$$

Now we successfully extend the original formula to a fronter layer. By repeating this process, we can approximate $\delta f_r$ by the gradient from every layer. Our original goal in Equation 5 is the 1-norm of $\delta f_r$. However, it is difficult to approximate via gradient. Instead, we calculate the sum of the square of each element in $\delta f_r$, which is highly positively correlated with its 1-norm. Based on the first-order derivative, we approximate it as:

$$(||\delta f_r||_2)^2 \approx f_r \cdot \delta f_r, \qquad (16)$$

where $\cdot$ is the dot product. Based on it and by extending Equation 15 to all previous layers, we derive:

$$F_{g_\theta}(v_r) \approx f_r\{\sum_{i=0}^{L-1}[(s(1 - \frac{d}{(N-1)(d+\beta)}))^{(L-1-i)} \frac{d_r}{d_r + \beta}$$
$$||\frac{\partial f_r}{\partial h_r^{(i)}} \circ h_r^{(i)}||_2] + k_3 \cdot L \cdot \delta Topo_r\}. \qquad (17)$$

In the formula, $h_i^{(0)}$ is the input feature of $v_i$. Since our derivation is from the back layer to the front layer, approximation error might accumulate. To eliminate this issue, we add an additional decay term $s$ to reduce the weight of fronter layers. $s$ usually falls in [0.9, 1.0]. Since each layer generates a $\delta Topo_r$ term, we multiply it by the number of layers $L$.

However, Equation 17 is still not efficient. It needs to backpropagate $f_r$ to acquire the approximation for node $v_r$, but we want to simultaneously generate the approximation results for all nodes. In the standard way, when we are backpropagating $f_r$, we set the loss of every node $v_i \in V$ as $f_r$, so that we can accurately get $\frac{\partial f_r}{\partial h_r^{(i)}}$. To upgrade the efficiency, We relax this restriction and set the loss of node $v_i \in V$ as $f_i$, allowing for each node to backpropagate a different loss. In this way, we can backpropagate them simultaneously. When we are approximating the influence of removing node $v_r$, we not only base on $f_r$ but also on $f_i, i \neq r$, so it downgrades the performance. However, $f_r$ still has a dominant influence on the gradient of $v_r$'s hidden representations, because self-loop and residual connections are stronger than normal edges. The experimental results show a satisfactory performance, so the accuracy drop is tolerable, with a huge gain in time efficiency. In this way, we can generate the approximation for all nodes simultaneously. It only takes a few seconds to complete the computation.

**Table 1: Complexity Comparision.**

| Type | Time | Space |
|---|---|---|
| brute-force | $O(LN^2h^2 + LNMh)$ | $O(M + Lh^2 + LNh)$ |
| NORA | $O(LNh^2 + LMh)$ | $O(M + Lh^2 + LNh)$ |

The approximation of the link prediction task is similar. We just replace $f_r$ with the sum of edge predictions which are not connected with $v_r$. The other processes during derivation are the same.

## 4.4 Complexity Analysis

Here we analyze the time and space complexity of the ground truth and the proposed method. We use $N$ to denote the number of nodes, $M$ to denote the number of edges, $L$ to represent the number of the GNN's layers, $h$ to represent the hidden size of the GNN model, and $d$ to represent the average node degree. In most cases, the adjacency matrix is sparsely stored, and in this situation, according to Paper [4], the time complexity of the forward propagation or backpropagation of a common message-passing GNN model is $O(LNh^2 + LMh)$, and the space complexity is $O(M + Lh^2 + LNh)$. Based on it, we list the time and space complexities in Table 1.

We list the detailed computation of these time and space complexity in the appendix A.2. As shown in Table 1, NORA cost significantly less time than the brute-force method, and basically the same space complexity as the brute-force method. Therefore, it is generalizable to very large real-world graphs when considering time. For example, it takes about 41 hours to generate the ground truth influence scores for DrGAT model on the ogbn-arxiv dataset, but it only takes a few seconds by NORA. When considering space, since they have the same space complexity as the GNN model, the bottleneck is the GNN's space consumption.

## 5 EXPERIMENTS

### 5.1 Baseline Adaption

Since there is no mature baseline for this new problem we propose, we adapt two methods from graph counterfactual explanation as baselines.

**CF-GNNExplainer.** CF-GNNExplainer [31] is a famous graph counterfactual explanation method. Its basic idea is to multiply the adjacency matrix with a mask matrix. It optimizes the mask matrix to drive the GNN prediction away from its original prediction. After training, a smaller element in the mask matrix indicates a more influential edge. We adapt it to evaluate node influence. We optimize a node mask $M \in R^{|V|}$, and its elements are limited in the range [0, 1]. In every GNN layer, we multiply node embeddings by $M$ before the message passing. After training, we evaluate influence as the distance between node mask and 1. Following CF-GNNExplainer, our loss function consists of a prediction loss term that drives the new prediction away from the original prediction and a regularization term that drives the value in the mask to be close to 1 (otherwise removing all nodes might be the best solution). The loss function is:

$$Loss = -\sum_{i=1}^{N} ||g_\theta(V, E)_i - g_\theta(V; E \circ M)_i||_1 + ||M||_1, \quad (18)$$

**Table 2: Dataset statistics.**

| Dataset | #Nodes | #Edges | #Features | #Classes | Homo/Hetero |
|---|---|---|---|---|---|
| Cora | 2,708 | 5,429 | 1,433 | 7 | homogeneous |
| CiteSeer | 3,327 | 4,732 | 3,703 | 6 | homogeneous |
| PubMed | 19,717 | 44,338 | 500 | 3 | homogeneous |
| ogbn-arxiv | 169,343 | 1,166,243 | 128 | 40 | homogeneous |
| P50 | 5,435 | 1,593,721 | - | 2 | heterogeneous |
| P_20_50 | 12,103 | 1,976,985 | - | 2 | heterogeneous |

**LARA.** LARA [41] is a recent work that greatly improves scalability by applying a GCN model to predict the edge influence. The GCN model generates a source embedding, $p_i$, and a target embedding, $t_i$ for every node $v_i \in V$. It predicts the influence of $v_i$ on $v_j$ by $p_i \cdot t_j$, where $\cdot$ is the dot product. We approximate the influence of node removal as the sum of its influence on its neighbors:

$$F_{g_\theta}(v_r) \approx \sum_{i \in N(r)} p_r \cdot t_i. \quad (19)$$

Besides, we also try to directly predict the node influence score with the GCN model, but it is not as effective as first generating node embeddings and calculating link influence.

## 5.2 Experiment Settings

*Datasets.* To comprehensively evaluate NORA in different scenarios, we conduct experiments on six datasets and two tasks. The datasets include four widely-applied benchmark citation networks (Cora, CiteSeer, and PubMed [42], and ogbn-arxiv [15]) and two social networks. Nodes on the four citation networks are papers, and undirected edges represent citations. The original task is to predict the research field of each paper. We also add a link prediction task to verify NORA's capacity in different settings. We follow the same data split ratio as the original link prediction task on the two social networks. The two social networks are heterogeneous Twitter datasets constructed by a previous study [56]. Nodes are users, and directed edges represent one of five Twitter actions or their counterparts (e.g., be followed): follow, retweet, like, reply, and mention. It originally contains two tasks. The first task is to predict the political leaning of each user. The second task is to predict whether there is a specific type of link from one user to another. Table 2 lists the dataset statistics.

An issue is that the trained GNN model is biased to the training-set nodes/edges. To fairly evaluate the influence of every node, we run each experiment 5 times and cycle the data split of nodes and edges by 20% per time, giving every node an equal chance to show up in training, validation, or test sets. For the link prediction task, we also cycle the link data split. After evaluation, we take the mean of the 5 results as the node influence score.

*GNN Models.* We select representative GNN models. On the citation datasets, we use three commonly used GNNs, GCN [21], Graph-SAGE [12], and GAT [49]. As the ogbn-arxiv dataset is a heated benchmark, we use the SOTA model on its leaderboard at the time we started this project, DrGAT [65], to replace the vanilla GAT. Dr-GAT is an improved variant of GAT, which is further equipped with a dimensional reweighting mechanism. Since the Twitter datasets

**Table 3: Approximation method performance and efficiency. We use GCN for Cora, CiteSeer, PubMed, and ogbn-arxiv datasets, and TIMME model for P50 and P_20_50.**

| Dataset | Method | Node Classification | | | | | Link Prediction | | | | |
|---|---|---|---|---|---|---|---|---|---|---|---|
| | | top-1 | top-5% | top-10% | Corr | Time | top-1 | top-5% | top-10% | Corr | Time |
| Cora | Betweenness | 100.0% | 74.6% | 72.9% | 0.763 | 26s | 100.0% | 79.0% | 78.4% | 0.864 | 26s |
| | CF-GNNExplainer' | 65.5% | 58.6% | 63.0% | 0.567 | 10s | 5.8% | 21.5% | 29.7% | 0.052 | 5.5s |
| | LARA-N | 100.0% | 90.2% | 89.4% | 0.815 | 4.7s | 100.0% | 92.6% | 89.9% | 0.770 | 4.6s |
| | LARA-E | 100.0% | 90.5% | 89.7% | 0.831 | 7.5s | 100.0% | 93.9% | 90.4% | 0.878 | 6.2s |
| | NORA | 88.8% | 92.6% | 91.9% | **0.884** | 11s | 100.0% | 91.0% | 89.0% | **0.907** | 13s |
| CiteSeer | Betweenness | 28.1% | 76.1% | 76.4% | 0.630 | 26s | 17.7% | 76.5% | 80.4% | 0.591 | 26s |
| | CF-GNNExplainer' | 78.0% | 37.4% | 38.6% | 0.478 | 9.3s | 1.8% | 24.4% | 31.4% | 0.018 | 6.5s |
| | LARA-N | 100.0% | 91.2% | 88.6% | 0.797 | 6.6s | 100.0% | 94.5% | 90.4% | 0.718 | 7.0s |
| | LARA-E | 100.0% | 89.8% | 85.7% | 0.812 | 6.2s | 100.0% | 96.0% | 94.7% | **0.917** | 7.4s |
| | NORA | 100.0% | 83.9% | 86.6% | **0.833** | 14s | 100.0% | 95.3% | 94.1% | 0.822 | 14s |
| PubMed | Betweenness | 63.3% | 76.8% | 85.4% | 0.528 | 42min | 66.4% | 80.1% | 86.3% | 0.569 | 42min |
| | CF-GNNExplainer' | 31.3% | 71.4% | 70.8% | 0.509 | 9.1s | 75.1% | 20.3% | 23.3% | 0.230 | 6.7s |
| | LARA-N | 79.6% | 90.2% | 91.0% | 0.799 | 3.8s | 39.1% | 88.7% | 93.1% | 0.837 | 4.6s |
| | LARA-E | 79.6% | 91.5% | 92.8% | **0.836** | 7.5s | 76.4% | 96.7% | 97.4% | **0.923** | 5.5s |
| | NORA | 51.1% | 83.2% | 88.6% | 0.745 | 19s | 100.0% | 89.1% | 91.4% | 0.873 | 22s |
| ogbn-arxiv | Betweenness | 100.0% | 74.4% | 77.9% | 0.782 | ≈140h | 100.0% | 75.8% | 78.9% | 0.786 | ≈140h |
| | CF-GNNExplainer' | 66.4% | 24.8% | 32.1% | 0.666 | 19s | 0.1% | 14.7% | 21.6% | 0.213 | 15s |
| | LARA-N | 100.0% | 86.0% | 83.5% | 0.595 | 9.1s | 100.0% | 91.5% | 89.4% | 0.559 | 21s |
| | LARA-E | 77.4% | 53.0% | 55.1% | 0.506 | 21s | 100.0% | 64.4% | 65.3% | 0.758 | 39s |
| | NORA | 77.4% | 86.5% | 86.1% | **0.900** | 35s | 100.0% | 95.6% | 94.2% | **0.997** | 31s |
| P50 | Betweenness | 100.0% | 83.6% | 91.8% | 0.643 | ≈6h | 100.0% | 72.3% | 86.2% | 0.644 | ≈6h |
| | CF-GNNExplainer' | 100.0% | 17.1% | 16.0% | 0.811 | 34s | 73.1% | 95.8% | 74.5% | 0.666 | 10min |
| | LARA-N | 100.0% | 89.4% | 92.1% | 0.435 | 10s | 100.0% | 81.1% | 88.2% | 0.540 | 59s |
| | LARA-E | 100.0% | 90.2% | 86.3% | 0.877 | 23s | 100.0% | 88.2% | 90.2% | 0.862 | 68s |
| | NORA | 100.0% | 98.7% | 98.5% | **0.956** | 19s | 100.0% | 92.3% | 91.3% | **0.943** | 24s |
| P_20_50 | Betweenness | 98.3% | 88.5% | 93.5% | 0.707 | ≈14h | 100.0% | 89.5% | 92.4% | 0.838 | ≈14h |
| | CF-GNNExplainer' | 66.9% | 62.5% | 57.4% | 0.612 | 76s | 100.0% | 21.6% | 21.9% | 0.789 | 15min |
| | LARA-N | 98.3% | 83.6% | 91.9% | 0.556 | 13s | 100.0% | 88.4% | 92.4% | 0.549 | 71s |
| | LARA-E | 98.3% | 93.7% | 93.1% | 0.968 | 25s | 100.0% | 94.2% | 93.3% | 0.968 | 84s |
| | NORA | 100.0% | 98.7% | 96.7% | **0.979** | 37s | 100.0% | 95.5% | 95.4% | **0.984** | 42s |

are heterogeneous, GCN, GraphSAGE and GAT are no longer useful, we use TIMME model, the GNN proposed in the same paper as the datasets [56]. It tackles the challenges on the Twitter datasets, e.g., sparse features, sparse labels, and heterogeneity.

*Evaluation Metrics.* We compare NORA against the baseline methods introduced above. In the following tables, "Betweenness" denotes the betweenness centrality; "CF-GNNExplainer'" is our adaption of CF-GNNExplainer. Among the adaptions of LARA, "N" and "E" represent the node-version and edge-version adaptions. We use two metrics to evaluate the similarity between approximation results and the ground truth. The first one is the top-k score, which is the sum of the influence score of the top k nodes ranked by the approximation method divided by that ranked by the ground truth. We evaluate top 1, top 5%, and top 10% nodes. The second metric is the Pearson correlation coefficient between the ground truth influence score and the approximated one.

*Hyper-parameters.* We keep the hyper-parameters for DrGAT and TIMME models the same as their original settings since they are already carefully tuned. We search for the best hyper-parameters for GCN, GraphSAGE, and GAT models. We also tune the hyper-parameters of each approximation method for each dataset and model. We list the hyper-parameter details in the appendix.

## 5.3 Performance Comparison

The main results of the compared methods are recorded in Table 3. We evaluate the approximation performance of the GCN model on each citation dataset, since GCN is one of the most commonly used GNN models. Since GCN is not applicable to the heterogeneous graph, we use TIMME model on the Twiter datasets. Table 4 shows the results of more GNN models on the node classification task on the four citation networks. In the two tables, NORA outperforms the baseline methods. The betweenness centrality can not take node attributes into consideration. The CF-GNNExplaner' method

**Table 4: Further Verification on More GNNs on the node classification task.**

| Dataset | GNN | Method | top-5% | top-10% | Corr |
|---------|-----|--------|--------|---------|------|
| Cora | GraphSAGE | LARA-E | 84.3% | 77.9% | 0.819 |
| | | NORA | 85.8% | 82.0% | **0.839** |
| | GAT | LARA-E | 84.8% | 83.6% | **0.792** |
| | | NORA | 76.7% | 78.3% | 0.774 |
| CiteSeer | GraphSAGE | LARA-E | 73.4% | 72.6% | 0.714 |
| | | NORA | 84.1% | 86.0% | **0.799** |
| | GAT | LARA-E | 80.7% | 76.9% | **0.782** |
| | | NORA | 79.5% | 78.3% | 0.746 |
| PubMed | GraphSAGE | LARA-E | 85.1% | 90.3% | 0.785 |
| | | NORA | 84.9% | 91.0% | **0.792** |
| | GAT | LARA-E | 84.3% | 87.6% | 0.794 |
| | | NORA | 92.1% | 93.7% | **0.915** |
| ogbn-arxiv | GraphSAGE | LARA-E | 39.6% | 43.0% | -0.007 |
| | | NORA | 92.6% | 90.8% | **0.961** |
| | DrGAT | LARA-E | 97.3% | 97.7% | 0.895 |
| | | NORA | 98.1% | 98.4% | **0.924** |

is useful in its original design, which is to analyze the influence on a single target node. However, when it considers all nodes or all edges, different nodes/edges might pick different influential nodes w.r.t. them, and thus the large mask is difficult to optimize. The LARA adaptions work best among the baselines, but they require a lot of labels, which must be generated by the time-consuming ground truth method. The original paper proposes a neighborhood sampling strategy to improve efficiency since it only targets one node, but it is not applicable in our scenario.

When comparing the efficiency, CF-GNNExplainer', LARA, and NORA are similar on small graphs. However, when we increase the graph size, NORA remains the most stable efficiency. Besides, LARA requires labeling of the ground truth to train the model. The time in the table does not include the labeling time, but it actually takes a lot of time. For example, it takes about 41 hours to generate the ground truth influence scores for the DrGAT model on the ogbn-arxiv dataset. If LARA requires 20% labeled data to train, it still needs about 8 hours. Calculating the betweenness centrality takes the longest time, since it traverses the shortest paths on the graph. For the ogbn-arxiv dataset, we only sample 10000 nodes to run the algorithm, and we approximate that it takes about 140 hours according to its time complexity.

### 5.4 Stability of The Proposed Influence Score

As the novel node-removal approach provides a new perspective of evaluating node influence, we want to examine whether the real influence of node removal generated by the brute-force method is stable across different GNNs and different hyper-parameters. We conduct experiments on the four citation datasets: Cora, CiteSeer, PubMed, and ogbn-arxiv. We use the same models as above. We change a sensitive hyper-parameter, hidden size, to evaluate the results' stability. For each model, we use three different hidden sizes: 128, 256, and 512, except for DrGAT on ogbn-arxiv, which only uses 128 and 256 due to memory limitation. For each model and each dataset, we traverse each two-hidden-size pair and calculate the

**Table 5: Stability results. The three column named by a GNN model shows the correlation coefficient of different results generated by the same GNN with different hidden sizes. The rightmost column means the correlation coefficient of different results generated by different GNN models.**

| Dataset | GCN | GraphSAGE | GAT/DrGAT | Inter-model |
|---------|-----|-----------|-----------|-------------|
| Cora | 0.9956 | 0.9857 | 0.9393 | 0.8765 |
| CiteSeer | 0.9968 | 0.9931 | 0.9585 | 0.8167 |
| PubMed | 0.9970 | 0.9963 | 0.9451 | 0.8372 |
| ogbn-arxiv | 0.9984 | 0.9979 | 0.9914 | 0.9557 |

Pearson correlation coefficient of each pair's results, and we report the mean of them. For each hidden size and each dataset, we also traverse each two-GNN pair and calculate the Pearson correlation coefficient of each pair's results, and we calculate the mean of them. Then, we further calculate the mean of the different hidden sizes' results ("Inter-model"). We list the results in Table 5.

From the results, we can observe that the performance generated by different hidden sizes are very similar. It indicates that the node-removal approach is stable across different hyper-parameters. Results generated by different GNN models are also quite similar. Nevertheless, it is not as similar as that of different hidden sizes. It indicates that the influence of node removal is still dependent on the specific GNN model.

## 6 CONCLUSION

It is important to study node influence and identify influential nodes on a graph. Existing approaches that capture node influence typically focus on how a node functions given its existence, but they ignore the node-removal perspective. We step into this important yet neglected perspective, which could provide a new perspective on node influence and benefit real-world applications. We use graph neural network (GNN) models as a surrogate to learning the underlying propagation patterns on a graph. We formalize the problem by removing a node, re-applying a trained GNN model, and using the output change to measure the influence.

For detecting the influence of node removal for each node, the ground-truth method is the brute-force algorithm, which is reliable but low in efficiency. To overcome this defect, we analyze how GNN's prediction changes when a node is removed and approximate it with gradient information. We propose **NO**de-**R**emoval-based f**A**st GNN inference (**NORA**). It can efficiently approximate such change in GNN's prediction for all nodes by one forward propagation and one backpropagation. As we are studying a new problem without mature baselines, we also adapt two methods from graph counterfactual explanation as baseline methods for comparison. We conduct extensive experiments on six networks and demonstrate NORA's effectiveness. We also verify the transferability of the node influence score across different models, which indicates that it is a stable indicator of node influence. This paper mainly focuses on the approximation and the influence of node removal. We hope this work can opens up an inspirational new perspective. In the future work, we would extend our proposed NORA to a broader line of research fields such as graph-level analysis, molecular property prediction and link prediction.

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

# A SUPPLEMENTARY DISCUSSIONS OF METHODS

Here we provide the supplementary discussions for the "Methods" section.

## A.1 Derivation of NORA

We first focus on the first term. To begin with, we get the following formula based on the chain rule:

$$\frac{\partial f_r}{\partial h_j^{(L-1)}} = \sum_{i \in N(j)} \left( \frac{\partial f_r}{\partial h_i^{(L)}} \frac{\partial h_i^{(L)}}{\partial h_j^{(L-1)}} \right) + \frac{\partial f_r}{\partial h_j^{(L)}} \frac{\partial h_j^{(L)}}{\partial h_j^{(L-1)}} \quad (20)$$

To simplify its form, we need to approximate the ratio of $\frac{\partial f_r}{\partial h_r^{(l)}} \frac{\partial h_r^{(l)}}{\partial h_j^{(l-1)}}$ to $\frac{\partial f_r}{\partial h_j^{(l-1)}}$. We use $d$ to represent the average degree. If $v_j$ is not $v_r$'s neighbor, the ratio is zero. If they are neighbors, which is of probability $d/(N-1)$, we assume that every neighbor of $v_j$ contribute equally to $\frac{\partial f_r}{\partial h_j^{(l-1)}}$. And we approximate that $v_j$ itself contributes $\beta$ times as its neighbors to the derivative. If we use $d$ to represent the average degree, then for $v_j$ being $v_r$'s neighbor, the ratio of $\frac{\partial f_r}{\partial h_r^{(l)}} \frac{\partial h_r^{(l)}}{\partial h_j^{(l-1)}}$ to $\frac{\partial f}{\partial h_j^{(l-1)}}$ can be approximated as $\frac{1}{d+\beta}$. Further, we assume that every node $v_j$ functions equally, then we acquire Equation 21.

$$\sum_{j \neq r} \left( \sum_{i \in N(j), i \neq r} \left( \frac{\partial f_r}{\partial h_i^{(l)}} \frac{\partial h_i^{(l)}}{\partial h_j^{(l-1)}} \right) \delta h_j^{(l-1)} \right)$$

$$= \sum_{j \neq r} \left( \frac{\partial f_r}{\partial h_j^{(l-1)}} - \frac{\partial f_r}{\partial h_r^{(l)}} \frac{\partial h_r^{(l)}}{\partial h_j^{(l-1)}} \right) \delta h_j^{(l-1)} \quad (21)$$

$$\approx \left( \sum_{j \neq r} \frac{\partial f_r}{\partial h_j^{(l-1)}} \delta h_j^{(l-1)} \right) \left( 1 - \frac{d}{N-1} \frac{1}{d+\beta} \right) \quad (22)$$

As shown in the "Methods" section, we derive NORA's approximation formula layer by layer and from back to front. We explained the approximation form on one GNN layer in the "Methods" section. Here we show how we derive the final formula, Equation 17. At first, we start our approximation from the output layer. Let's assume we are removing node $v_r$ and the GNN model has $L$ layers. The approximation begins with:

$$F_{g_\theta}(v_r) \approx \delta f_r \approx \sum_{i \neq r} \frac{\partial f_r}{\partial h_i^{(L)}} \delta h_i^{(L)} \quad (23)$$

Then with the approximation of the three terms introduced in the "Methods" section, we can transform the above formula to:

$$F_{g_\theta}(v_r) \approx \left( \sum_{j \neq r} \frac{\partial f_r}{\partial h_j^{(L-1)}} \delta h_j^{(L-1)} \right) \left( 1 - \frac{d}{N-1} \frac{1}{d+\beta} \right)$$

$$+ \delta Topo_r + \frac{|\hat{N}(r)|}{|\hat{N}(r)| + \beta} \left\| \frac{\partial f_r}{\partial h_r^{(L-1)}} \circ h_r^{(l-1)} \right\|_2$$

We rewrite $j$ with $i$, and then we get:

$$F_{g_\theta}(v_r) \approx \left( \sum_{i \neq r} \frac{\partial f_r}{\partial h_i^{(L-1)}} \delta h_i^{(L-1)} \right) \left( 1 - \frac{d}{N-1} \frac{1}{d+\beta} \right)$$

$$+ \delta Topo_r + \frac{|\hat{N}(r)|}{|\hat{N}(r)| + \beta} \left\| \frac{\partial f_r}{\partial h_r^{(L-1)}} \circ h_r^{(L-1)} \right\|_2 \quad (24)$$

The first part of the formula has a similar algebraic form as Equation 23. We approximate the term $\sum_{i \neq r} \left( \frac{\partial f_r}{\partial h_i^{(L-1)}} \delta h_i^{(L-1)} \right)$ in the same way, so it extends the formula to previous layers. As the approximation error might accumulate through layers, we multiply the term by an extra decay weight $s \in [0, 1]$ to mitigate the contribution of former layers:

$$F_{g_\theta}(v_r) \approx \left( \sum_{i \neq r} \frac{\partial f_r}{\partial h_i^{(L-1)}} \delta h_i^{(L-1)} \right) s \left( 1 - \frac{d}{N-1} \frac{1}{d+\beta} \right)$$

$$+ \delta Topo_r + \frac{|\hat{N}(r)|}{|\hat{N}(r)| + \beta} \left\| \frac{\partial f_r}{\partial h_r^{(L-1)}} \circ h_r^{(L-1)} \right\|_2 \quad (25)$$

We expand the formula to previous layers and approximate previous layers similarly. When we reach the input layer, we get:

$$F_{g_\theta}(v_r) \approx \sum_{i=1}^{L-1} \left( s \left( 1 - \frac{d}{N-1} \frac{1}{d+\beta} \right) \right)^{(L-1-i)} \cdot$$

$$\left( \frac{|\hat{N}(k)|}{|\hat{N}(k)| + \beta} \cdot \left\| \frac{\partial f_r}{\partial h_r^{(i)}} \circ h_r^{(i)} \right\|_2 + \delta Topo_r \right)$$

$$+ \left( \sum_{i \neq r} \frac{\partial f_r}{\partial h_i^{(0)}} \delta h_i^{(0)} \right) \left( s \left( 1 - \frac{d}{N-1} \frac{1}{d+\beta} \right) \right)^L$$

$$+ \left( s \left( 1 - \frac{d}{N-1} \frac{1}{d+\beta} \right) \right)^{(L-1)} \left( \delta Topo_r + \frac{|\hat{N}(r)|}{|\hat{N}(r)| + \beta} \left\| \frac{\partial f_r}{\partial h_r^{(0)}} \circ h_r^{(0)} \right\|_2 \right) \quad (26)$$

In the formula, $h_i^{(0)}$ is the input feature of $v_i$. It won't change, so $\delta h_i^{(0)} = 0$. Since $\delta Topo_r$ is the same in every layer, we extract it from the summation and assign it a weight $k_3$. Then we can get the final formula:

$$F_{g_\theta}(v_r) \approx \sum_{i=0}^{L-1} \left( s \left( 1 - \frac{d}{N-1} \frac{1}{d+\beta} \right) \right)^{(L-1-i)} \cdot \frac{|\hat{N}(k)|}{|\hat{N}(k)| + \beta} \cdot$$

$$\left\| \frac{\partial f_r}{\partial h_r^{(i)}} \circ h_r^{(i)} \right\|_2 + k_3 \cdot L \cdot \delta Topo_r \quad (27)$$

## A.2 Time and Space Complexity

Here we make a detailed analysis of the methods' time and space complexity. $N$ denotes the number of nodes, $M$ denotes the number of edges, $L$ represents the number of the GNN's layers, $h$ represents the hidden size of the GNN model, and $d$ is the average node degree. In most cases, the adjacency matrix is sparsely stored, and in this situation, according to Paper [4], the time complexity of the forward propagation or backpropagation of a common message-passing GNN model is $O(LNh^2 + LMh)$, and the space complexity is $O(M +$

$Lh^2 + LNh$). Here we list the time complexity of the ground truth and NORA:

- Brute-force method (ground truth): it removes the nodes one by one and does the forward propagation. The average time complexity of removing a node is $O(d)$. Therefore, the total time complexity is $O(N(LNh^2 + LMh + d)) = O(LN^2h^2 + LNMh)$.
- NORA: NORA first does a forward propagation and a back-propagation, which costs $O(LNh^2 + LMh)$. In NORA's formula (Formula 14 in Section 4.1), the part before the plus sign takes $O(h)$ to calculate the 2-norm of the dot product for each layer and each node, so it totally takes $O(LNh)$. The part after the plus sign needs to calculate $\delta topo_r$. Calculating $\delta topo_r$ makes two aggregations of neighbor information, each of which takes $O(M)$ for all nodes, so it takes $O(2M) = O(M)$. Therefore, NORA totally takes $O(LNh^2 + LMh + LNh + M) = O(LNh^2 + LMh)$, which is the same as GNN's propagation itself.

Here we list the space complexity of the ground truth and NORA apart from the space complexity of GNN itself, $O(M + Lh^2 + LNh)$.

- Brute-force method: it additionally stores a modified graph, which costs $O(M)$..
- NORA: it additionally stores the gradients of every hidden layer and some middle results, which costs $O(M + LNh)$.

None of these additional space complexity is comparable with the space complexity of the GNN model, so their space complexity is still $O(M + Lh^2 + LNh)$.

## B  HYPER-PARAMETERS

On ogbn-arxiv, P_50, and P_20_50, we use their original data split ratio. On Cora, CiteSeer, and PubMed, the majority of nodes are not in any of the training, validation, or test set, so we change the data split ratio to 5:3:2 to cover all nodes.

### B.1  Hyper-Parameters of GNN Models

We have used five GNN models: GCN, GraphSAGE, GAT, DrGAT, and TIMME. As GCN, GraphSAGE, and GAT are widely-used GNNs on various datasets, we tune their hyper-parameters and choose a well-performing setting. For DrGAT on ogbn-arxiv and TIMME on the two Twitter datasets, we keep them the same as their original choices. Please refer to DrGAT's implementation repository [1] and TIMME's official repository [2] for more details. On Cora, CiteSeer, and PubMed datasets, we adapt GCN, GraphSAGE, and GAT models from PyG. On ogbn-arxiv dataset, we adapt GCN and GraphSAGE models from the implementation [3] of OGB team. We adapt DrGAT from its implementation repository. On the two Twitter datasets, we adapt TIMME from its official repository. TIMME consists of multiple tasks, including a node classification task and some auxiliary edge prediction tasks, among which we only care about the node classification task's output.

Here we describe our hyper-parameter settings of GCN, Graph-SAGE, and GAT. They have two layers when operating on Cora,

CiteSeer, or PubMed, and three layers when operating on ogbn-arxiv. On Cora, CiteSeer, or PubMed, they are trained with the early-stopping mechanism. On ogbn-arxiv, GCN and GraphSAGE are trained with fixed 300 epochs. We save the model at the epoch where the validation performance reaches the highest. Later we use that saved model to generate the influence of node removal. The learning rate is set to $1e - 2$, except for $3e - 3$ when training GAT on PubMed. Other hyper-parameters are listed in Table 6.

**Table 6: Hyper-parameters of GNN models. "#epoch" and "patience" are the maximum number of epochs and the patience used for early-stopping. "wd" is the weight decay.**

| Dataset | GNN | wd | hidden | dropout | #epoch | patience |
|---|---|---|---|---|---|---|
| Cora | GCN | 1e-4 | 1024 | 0.6 | 50 | 20 |
|  | GraphSAGE | 1e-4 | 256 | 0.9 | 100 | 50 |
|  | GAT | 3e-5 | 1024 | 0.5 | 200 | 100 |
| CiteSeer | GCN | 3e-4 | 1024 | 0.5 | 200 | 50 |
|  | GraphSAGE | 1e-4 | 128 | 0.9 | 100 | 50 |
|  | GAT | 1e-4 | 256 | 0.5 | 150 | 70 |
| PubMed | GCN | 2e-4 | 1024 | 0.5 | 50 | 20 |
|  | GraphSAGE | 1e-4 | 256 | 0.5 | 150 | 70 |
|  | GAT | 4e-4 | 1024 | 0.3 | 150 | 70 |
| ogbn-arxiv | GCN | 0 | 256 | 0.5 | 300 | - |
|  | GraphSAGE | 0 | 256 | 0.5 | 300 | - |

### B.2  Hyper-Parameters of NORA

We need to slightly modify the NORA algorithm on the Twitter datasets. Since the Twitter datasets are directed graphs and each edge has a reversed counterpart, we change $N(i)$ and $\hat{N}(i)$ into $v_i$'s in-neighbors or out-neighbors, instead of containing each neighbor twice. Similarly, we need to change $d$ to the average in-degree or average out-degree.

NORA has five hyper-parameters in Equation 17: $k_1$, $k_2$, $k_3$, $s$ and $\beta$. We tune them for each dataset and model. Usually, the best hyper-parameter setting for one metric is not the best for another. We consider all the metrics when selecting the hyper-parameters. We also report the highest Pearson correlation coefficient results with hyper-parameters to maximize this metric. In the experiments of NORA-t and NORA-n, we use the same hyper-parameters for NORA on the same GNN model and dataset. In the experiments that don't consider Precision@k%, we only consider Pearson correlation coefficient metric, so we use the hyper-parameters that maximize Pearson correlation coefficient.

$k_1$, $k_2$, and $s$ are limited in range $[0, 1]$. $s$ is usually set to 0.95 or 1. $k_3$ differs greatly on different models and datasets, since the scale of NORA-t and NORA-n differs greatly in different situations. $\beta$ typically falls in $[2, 30]$. The hyper-parameters of NORA that only considers Pearson correlation coefficient are listed in Table ??. Experiments of deeper GNNs consider more than 3 layers. Other experiments only use 2 layers on Cora, CiteSeer, and PubMed; and 3 layers on ogbn-arxiv. The hyper-parameters that consider both Precision@k% and Pearson correlation coefficient are listed in Table 8.

---

[1]https://github.com/anonymousaabc/DRGCN
[2]https://github.com/PatriciaXiao/TIMME
[3]https://github.com/snap-stanford/ogb/tree/master/examples/nodeproppred/arxiv

**Table 8: Hyper-parameters of NORA that consider both Precision@k% and Pearson correlation coefficient.**

| Dataset | GNN | $k_1$ | $k_2$ | $k_3$ | $s$ | $\beta$ |
|---------|-----|-------|-------|-------|-----|---------|
| Cora | GCN | 0.9 | 0.5 | 20 | 0.95 | 6 |
| | GraphSAGE | 0.6 | 0.3 | 100 | 1.0 | 6 |
| | GAT | 0.6 | 0.6 | 50 | 1.0 | 2 |
| CiteSeer | GCN | 0.9 | 0.8 | 10 | 0.95 | 3 |
| | GraphSAGE | 1.0 | 1.0 | 70 | 1.0 | 4 |
| | GAT | 0.9 | 0.9 | 20 | 0.95 | 4 |
| PubMed | GCN | 0.4 | 1.0 | 500 | 0.95 | 25 |
| | GraphSAGE | 0.2 | 1.0 | 3000 | 0.95 | 20 |
| | GAT | 0.5 | 0.4 | 120 | 0.95 | 7 |
| ogbn-arxiv | GCN | 1.0 | 1.0 | 1.3e4 | 0.95 | 6 |
| | GraphSAGE | 1.0 | 1.0 | 2e4 | 0.95 | 6 |
| | DrGAT | 1.0 | 1.0 | 1e4 | 0.95 | 2 |
| P50 | TIMME | 0.05 | 0.07 | 7e4 | 1.0 | 3 |
| P_20_50 | TIMME | 0.1 | 0.1 | 3e4 | 0.95 | 4 |

