# OpenReview forum: "Evaluating Task-Specific Node Influence via Node-Removal-Based Fast Graph Neural Network Inference"
_ACM.org/TheWebConf/2024/Conference — TheWebConf24_

### Official Review · Reviewer_q6eb · 2023-11-07

**Novelty:** 6
**Technical Quality:** 4

**Review:**

This paper proposes a method to efficiently compute the influence of node removal for each node.
In particular, the proposed method reduces the running time from $O(N(LMh+LNh^2))$ to $O(LMh+LNh^2)$ by an approximation method.


Pros:\
S1: The paper is well-structured and effectively communicates its concepts and methodologies.\
S2: The technical approach is sound, and in several instances, the proposed method demonstrates superior performance to baseline approaches.\
S3: To my knowledge, this is the first study to successfully reduce the computational time complexity involved in determining the influence of node removal.

Cons:\
W1: The paper does not thoroughly investigate the approximation error. It lacks a theoretical error bound and does not include ample experimental data to empirically limit the error. Such information is crucial for evaluating the reliability of the method.\
W2: The differences between the NORA and LARA methods could be more distinctly delineated, providing readers with better guidance on selecting the most suitable approach for different contexts.

**Questions:**

Q1: What is the approximation error associated with the proposed method? Understanding whether the reduction in time complexity significantly impacts accuracy is important. A clear assessment of the proposed method's ability to maintain accuracy, compared to the brute force approach, would enhance the paper's contribution.\
Q2: How does the proposed method differ from LARA? Given that the proposed method does not outperform LARA in all cases, providing a detailed comparison of the two approaches could assist users in selecting the most suitable method for their specific scenarios.

**Reviewer Confidence:**

3: The reviewer is confident but not certain that the evaluation is correct

**Scope:**

4: The work is relevant to the Web and to the track, and is of broad interest to the community

---

### Official Review · Reviewer_JSFn · 2023-11-23

**Novelty:** 2
**Technical Quality:** 2

**Review:**

This paper designs an efficient method to explore the influence of the node, termed NORA. NORA uses the gradient of the network to approximate the noed-removal results.

**Questions:**

1. The introduction is confusing, lacking a clear explanation of the existing problems and motivation.

2. Inconsistency in the title, as the paper title does not match the registered title.

3. Figure 1 is perplexing; it is unclear what the figure intends to convey.

4. The idea of combining a pre-trained model with the model after removing nodes to understand node impact seems straightforward.

5. The experimental results in Table 3 seem to suggest that NORA is not competitive, especially in terms of time consumption.

6. Please have the author elaborate on the significance of studying the removal of a single node on model performance. Are there any practical applications for such a study?

**Reviewer Confidence:**

4: The reviewer is certain that the evaluation is correct and very familiar with the relevant literature

**Scope:**

3: The work is somewhat relevant to the Web and to the track, and is of narrow interest to a sub-community

---

### Official Review · Reviewer_NrUG · 2023-11-23

**Novelty:** 5
**Technical Quality:** 5

**Review:**

This paper introduces a practical problem of evaluating the influence of node existence in graph neural networks (GNNs). They propose an efficient algorithm called NORA that approximates each node's influence score by using the neural network's gradient. Experimental results demonstrate the effectiveness of their approach.

**Questions:**

1. The title in your submitted PDF differs from it in openreview submission.

2. Can you explain more about the performance variances between different datasets?

**Reviewer Confidence:**

2: The reviewer is willing to defend the evaluation, but it is likely that the reviewer did not understand parts of the paper

**Scope:**

3: The work is somewhat relevant to the Web and to the track, and is of narrow interest to a sub-community

---

### Official Review · Reviewer_GhZT · 2023-11-23

**Novelty:** 5
**Technical Quality:** 5

**Review:**

The topic of this paper is to identify influential nodes in graphs. The authors investigate a new yet practical problem of evaluating the influence of overall changes in the outputs of a trained GNN model caused by removing a node. They point out that adversarial attacks and counterfactual explanations cannot satisfy the needs due to their low efficiency. An efficient algorithm named NORA is proposed, which uses the gradient of neural networks to approximate the results of node removal. Experimental results on six datasets demonstrate its superiority.

**Strengths**
+ The experiments are extensive.
+ The code is available and the hyper-parameters are provided.
+ The research problem is interesting and meaningful.

**Weaknesses**
+ Only two tasks are conducted. I suggest verifying the methods on more graph tasks, such as node clustering.
+ In the formulas, the matrix needs to be presented as uppercase boldface, while the vector needs to be presented as lowercase boldface.
+ The efficiency of NORA seems limited. In Table 3, for node classification, the time taken by NORA is longer than LARA-E on 5 out of 6 datasets.
+ The comparison experiment only considers GCN and TIMME as the backbones.
+ Why did the authors keep the model parameters unchanged? I believe that in real-world scenarios, node removal would affect the training process of GNNs and the model parameters.

**Questions:**

See weaknesses.

**Reviewer Confidence:**

3: The reviewer is confident but not certain that the evaluation is correct

**Scope:**

3: The work is somewhat relevant to the Web and to the track, and is of narrow interest to a sub-community

---

### Official Review · Reviewer_KZG4 · 2023-11-23

**Novelty:** 5
**Technical Quality:** 5

**Review:**

This paper presents a novel approach to assess the influence of individual nodes in graph neural networks (GNNs). It addresses the challenge of measuring the impact on GNN outputs when a node is removed, a task that is traditionally time-consuming and inefficient. The authors introduce an algorithm named NORA (NOde-Removal-based fast GNN inference), which leverages neural network gradients to approximate node-removal effects. NORA enhances efficiency by requiring only one forward and one back propagation for calculating the influence scores of all nodes.

Strong points:
1. The proposed NORA streamlines the process of evaluating the influence of node removal in GNNs, which is a notable improvement over traditional methods.
2. NORA stands out for its computational efficiency. It requires only one forward and one back propagation to approximate the influence scores for all nodes in a network.
3. The paper's credibility is bolstered by extensive experiments conducted on six benchmark datasets.

Weak points:
1. The depth of the novelty might be questioned, as the concept fundamentally revolves around node influence measurement, a theme already prevalent in graph analysis literature. While the specific angle of node removal influence is new, it could be seen as an incremental advancement rather than a revolutionary one. The paper might have benefited from a more in-depth exploration of how this approach fundamentally changes or challenges existing paradigms in graph neural network research​​.
2. The introduction and related work sections could have been more expansive, providing a richer background into how this work fits into the broader landscape of graph neural network research. Additionally, the writing could have better highlighted the implications of this research beyond the technical contribution, such as its potential impact on real-world applications or how it advances our understanding of graph structured data​​.

**Questions:**

Please refer to the weaknesses pointed out in the Review.

**Ethics Review Description:**

NA.

**Reviewer Confidence:**

3: The reviewer is confident but not certain that the evaluation is correct

**Scope:**

4: The work is relevant to the Web and to the track, and is of broad interest to the community

---

### Official Review · Reviewer_ZaXS · 2023-11-30

**Novelty:** 4
**Technical Quality:** 6

**Review:**

This paper addresses a critical problem to approximate the influence of node-removal on graph neural networks. Instead of removing one node at a time and retrain the GNN, the proposed method approximates the ground-truth node-removal results uses the gradient of the graph neural networks, which is way more efficient. Empirical results show that the proposed method achieves a decent approximation of the ground-truth node-removal influence, which is both more effective and significantly more efficient than multiple baseline methods.

Pros:
* The problem of understanding node-removal influence on GNNs is critical, and this is one of the few methods that can approach this task practically (without retraining the GNN every time when removing a node).
* The paper presents both solid theoretical results and strong empirical results.

Cons:
* The main concern is that the paper misses an important reference: Chen et al., Characterizing the Influence of Graph Elements, ICLR 2023 (https://openreview.net/forum?id=51GXyzOKOp). This recent paper proposed a very similar idea, also using the gradients information to approximate the influence of node-removal, and they also presented theoretical bounds and strong empirical results. A comparison to this existing approach (both analytically and empirically) is crucial for the readers to assess the novelty and practical value of the proposed method.

In conclusion, this is an excellent paper addressing a critical and previously infeasible problem. However, the existence of a recent relevant work may have shadowed the novelty and value of the proposed work.

**Questions:**

* Please add a reference with Chen et al., articulating the difference between the two methods and comparing their empirical performance.

**Reviewer Confidence:**

4: The reviewer is certain that the evaluation is correct and very familiar with the relevant literature

**Scope:**

4: The work is relevant to the Web and to the track, and is of broad interest to the community

---

### Decision · Program_Chairs · 2024-01-22

**Decision:**

Accept

**Comment:**

Summary: An efficient way to approximate the influence of node-removal on GNNs using gradients.

 Strengths:
 + Good problem
 + Efficient and scalable method
 + Mild theoretical backing
 + No retraining needed
 + Solid experiments

 Weaknesses:
 - Novelty on the lower side
 - No approximation analysis (even for convex case)
 - Some overlap with ICLR 2023 paper

 Recommendation: Please address the comments in the reviews, especially the overlap with the ICLR 2023 paper and the additional graph tasks.